# Human Cytomegalovirus Reactivation During Lactation: Impact of Antibody Kinetics and Neutralization in Blood and Breast Milk

**DOI:** 10.3390/nu12020338

**Published:** 2020-01-28

**Authors:** Katrin Lazar, Tabea Rabe, Rangmar Goelz, Klaus Hamprecht

**Affiliations:** 1Institute for Medical Virology and Epidemiology of Viral Diseases, University Hospital Tuebingen, 72076 Tuebingen, Germany, Tabea.Rabe@med.uni-tuebingen.de (T.R.); 2Department of Neonatology, University Children’s Hospital Tuebingen, 72076 Tuebingen, Germany; Rangmar.Goelz@med.uni-tuebingen.de

**Keywords:** HCMV, milk whey, breastfeeding, IgG, p150, glycoprotein B, neutralization, recomLine blot

## Abstract

Human cytomegalovirus (HCMV) is shed into breast milk in nearly every seropositive woman during lactation. This reactivation shows mostly a self-limited, unimodal course. The dynamics and functional role of HCMV-specific-IgG in breast milk and in plasma during reactivation are unknown. Milk whey viral loads were monitored with real-time PCR in 18 HCMV-seropositive mothers over two months postpartum. HCMV-antibody binding assays (ECLIA) and antigen-specific immunoblotting were performed from plasma and corresponding milk samples. Epithelial-cell-specific neutralization was used to analyze functional antibodies in plasma- and whey-pools. Viral loads in milk whey showed unimodal courses in 15 of 18 mothers with peak viral loads around one month postpartum. HCMV-specific-IgG-antibodies increased significantly in plasma and milk whey during reactivation. The mean levels of plasma IgG were about 275-fold higher than in whey. Only antibodies against tegument protein p150 were continuously expressed in both compartments. Anti-glycoprotein-B1 IgG-antibodies were variably expressed in whey, but continuously in plasma. Neutralization assays showed 40-fold higher NT-50 values in plasma compared to whey at two months postpartum. During reactivation, HCMV-specific-IgG reactivities and neutralizing capacities are much lower in whey than in plasma. Therefore, their specific role in the decrease and discontinuation of virus-shedding in milk remains unclear.

## 1. Introduction

HCMV belongs to the β-herpesvirus family and plays a major role in immunosuppressed patients in the transplant setting and in the context of congenital and postnatal HCMV-infection [1]. Locally restricted HCMV-reactivation during lactation can be easily studied by viral DNA (DNAlactia), as well as by infectious virions isolated from breast milk in microculture (virolactia) [2]. The mechanism of postnatal HCMV-reactivation in the mammary gland without viral systemic infection (DNAemia) is still unknown. Virus-shedding into breast milk of immunocompetent healthy breastfeeding mothers occurs in nearly every seropositive mother at any time point during lactation, but usually ends 2–3 months after birth [3,4]. Transmission via breastfeeding can lead to severe symptomatic HCMV-infection in preterm infants, with increased risk with birth weight below 1500 g or less than 32 weeks of gestational age [2]. HCMV disease in the neonatal period is not defined in detail compared to the transplant setting [5]. A first case report on immunohistochemically proven manifestation of postnatally acquired HCMV colitis of a term infant was published recently [6]. Goelz et al. [7] could document volvulus as an additional severe HCMV-associated intestinal disease in preterm infants. 

Nevertheless, breast milk is considered the most important biofluid for protecting the infant’s health and supporting the maturation of its immune system [8,9]. Colostrum (up to one week postpartum (pp)) is considered especially important for newborns. Colostrum is followed by transient milk (2–4 weeks pp) and mature milk (more than 4 weeks pp). The World Health Organization (WHO) recommends exclusive breastfeeding for the first six months of life [10]. It was shown that respiratory and gastrointestinal infections decreased in infants exclusively fed with breast milk for 4 months pp [11]. Nutritive and immunologically important key components of human milk, e.g., lactoferrin, lysozyme, oligosaccharides, vitamins, immunoglobulins and cytokines, were reviewed recently by Wesolowska et al. [12] and Boix-Amorós et al. [13]. 

Antibodies are transferred transplacentally from mother to infant and also postnatally via breast milk. Postnatally acquired antibodies from breast milk are reported not to reach the infant’s circulation [14], but can still bind toxins, bacteria and other pathogens in the intestine [9,15]. The most abundant antibody in breast milk is soluble (s) IgA with up to 25% of the total protein concentration in colostrum [16], representing an important innate mucosal immune protection for the infant [17]. IgG-antibodies are the second most common antibody class in human milk [9], comprising about 2% of total protein in colostrum, which later declines in mature milk [15]. 

In the case of HCMV, the humoral immune protection of the infant can be provided by intrauterine transfer of anti-HCMV-specific antibodies and through their presence in breast milk. For example, a highly neutralizing antibody was found against the glycoprotein B (gB) [18]. One working hypothesis is that HCMV-specific antibodies are responsible for the self-limited HCMV-reactivation in the mammary gland. Moreover, neutralizing antibodies, especially against gB, should be detectable and might increase during lactation or play a role in HCMV-specific neutralization. The role of defined T cell subsets in immune control of HCMV-reactivation during lactation remains to be determined. To find a better understanding of these mechanisms, we inaugurated the BlooMil study in which we simultaneously evaluated blood and breast milk during the first 8 weeks postnatally at defined time ranges in a cohort of HCMV-seronegative and -seropositive breastfeeding mothers of mostly preterm infants.

## 2. Materials and Methods 

### 2.1. Study Design

The BlooMil study was performed in collaboration with the Department of Neonatology at the University Children’s Hospital Tuebingen. In total, 36 healthy and immunocompetent mothers with known HCMV-serostatus participated, 8 had to be excluded from the analysis due to lack of breast milk or transfer to a peripheral hospital. The remaining 28 mothers provided simultaneously sampled 9 mL specimens of EDTA blood and up to 80 mL of breast milk at four different time ranges (T) after birth: T1: 10–15, T2: 25–30, T3: 40–45, T4: 55–60 days pp. In total, 112 breast milk samples and 112 corresponding blood samples were collected. A total of 25 mothers had preterm infants with gestational ages ranging from 23 6/7 to 34 2/7 weeks and three had full term infants. Ten mothers were HCMV-seronegative and 18 were seropositive. As a part of the BlooMil study, four pairs of plasma and milk whey from each of these 18 seropositive mothers were longitudinally analyzed for HCMV-reactivation in breast milk, HCMV-specific antibodies and neutralization capacity in both plasma and whey. The mothers had all given their written informed consent. The study protocol was approved by the Clinical Ethics Committee at the University Hospital Tuebingen (project number: 804/2015BO2).

### 2.2. Breast Milk Preparation

Up to 80 mL of pumped human milk was collected at each time range of the BlooMil study. Breast milk was prepared as described by Hamprecht et al. [19], with the following modifications. The first centrifugation step was at 500g for 10 min at 4 °C. The creamy fat top layer was discarded and the whey was centrifuged again at 1780g for 10 min and filtrated using a pore size of 0.45 µm.

### 2.3. Viral Monitoring

DNA extraction from whey was done using the QIAcube DNA extractor and QIAamp Blood Mini Kit from Qiagen (Hilden, Germany). Nested HCMV IE1-Ex4 PCR from plasma and whey extracts have been described elsewhere [20,21]. With an organic extraction method, the limit of detection (LOD) was at 200 copies/mL [21]. Quantitative DNA levels were detected with real-time PCR (CMV R-gene® Kit, Biomérieux, Marcy-l’Étoile, France) of whey using Light Cycler 2.0 (Roche, Basel, Switzerland, LOD 600 copies/mL). 

### 2.4. HCMV-Specific-IgG Antibodies

Quantitative HCMV-specific-IgG was measured by electrochemiluminescence immunoassay ECLIA (ECLIA-Elecsys CMV-panel, using pp28, pp150, p52, p38 recombinant antigens for IgG (p150, p52 recombinant antigens for IgM): non-reactive ≤ 0.5 U/mL (< 0.7 COI), indeterminate = 0.5–1.0 U/mL (0.7–1.0 COI), reactive ≥1 U/mL (≥1 COI), cobas6000 module e601; Roche, Basel, Switzerland). For validation of whey, a spike and recovery experiment with the ECLIA test system was performed. Therefore, whey and the PBS control were spiked with a 100-fold diluted hyperimmunoglobulin (HIG: Cytotect®, Biotest® Pharma; charge B797033, 50 mg/mL) stock solution. The recovery rate in the spiked whey samples was 88.3%.

For the determination of antigen-specific IgG, immunoblotting with recombinant proteins was additionally performed using recomLine CMV test from Mikrogen Diagnostik (Neuried, Germany). For immunoblotting, whey was diluted two-fold. In contrast, plasma was diluted 100-fold as described by the manufacturer. IgG-antibody patterns were screened against the following six HCMV recombinant antigens: IE1, CM2, p150, p65, gB1 and gB2 [22]: Immediate early antigen 1 (IE1) is encoded by the gene UL123 and is an immediate early gene product transcribed during very early steps of HCMV gene expression regulating viral replication. CM2 is a fusion protein of UL57, a single-strand DNA-binding protein, and pUL44 is a processivity factor of viral polymerase. IgM-antibodies against CM2 were shown to be expressed during early HCMV primary infection [23,24]. pp150, encoded by UL32, is a highly immunogenic tegument phosphoprotein of HCMV [25]. pp65, encoded by UL83, is also a tegument phosphoprotein of the lower matrix and shows high response as T cell epitope. The glycoprotein gB, encoded by UL55, is detected by reactivity against gB1 and gB2 in the recomLine blot. gB1 antigen contains the amino acid positions 550 to 640 of gp58, identified as antigenic domain 1 (AD1), and amino acids 28 to 84 of gp116, identified as antigenic domain 2 (AD2). gB2 antigen is a fusion protein of AD2 from two strains AD169 and Towne [26]. IgG-antibodies against gB were shown to be highly neutralizing [18]. Scores of HCMV-specific-IgG reactivity were given on the y-axis, as described by Kagan et al. [27] and the number of study participants is shown on the x-axis. A validation experiment was performed, using HIG-spiked whey. Antibodies against all antigens were detectable without any influence on reactivity against the phosphoproteins p65 and p150. Only a very slight decrease in reactivity was observed against the recombinant antigens gB1 and gB2, IE1 and CM2. This did not interfere with our antibody analysis in milk whey.

### 2.5. Neutralization Assays

Neutralization (NT) assays using human retinal pigment epithelial cells (ARPE-19) were performed as described earlier by Maschmann et al. [28] and Schampera et al. [29]. As controls, HCMV-seronegative pools of plasma and whey from mothers of the BlooMil study were used. NT calibration in plasma or whey was done by dilutions of a HIG stock solution (Cytotect®) to concentrations of in vivo HCMV-IgG levels of the seropositive cohort: The mean ECLIA values of the seropositive plasma (720 U/mL) and whey samples (6.4 U/mL) were calibrated by predilution of Cytotect® (4725 U/mL) [29] in medium to both corresponding concentrations (1:6.5 and 1:700 respectively). Then, every control was measured in the same dilution steps as the HCMV-seropositive pools. For graphical display of NT-capacity, the HCMV-seronegative pools were used as 100% reference for each corresponding HCMV-seropositive pool dilution step. In general, a high NT-capacity corresponded to low HCMV immediate early antigen (IEA) plaque numbers.

### 2.6. Statistical Analysis

For statistical evaluations of HCMV-specific-IgG ECLIA kinetics, the Friedman test with Bonferroni correction was used in SPSS (version 25.0.0.1, IBM, Armonk, USA). A Wilcoxon matched-pairs signed rank test was performed using GraphPad Prism 8.1.0 (GraphPad Software Inc., San Diego, USA) to compare single time ranges with each other. The relation of plasma to milk whey HMCV-specific-IgG was evaluated with Spearman correlation and non-linear regression in GraphPad Prism. The Probit analysis was used for calculating the NT-50 values with corresponding mean values of the seronegative plasma and whey pools in SPSS (version 25.0.0.1, IBM, Armonk, USA) [29].

## 3. Results

### 3.1. HCMV-Reactivation in Breast Milk

Nearly all of the BlooMil study mothers reactivated the virus in milk whey (17 of 18, 94.4%) and showed an increase, reaching a peak level, followed by a decrease (unimodal course) of DNAlactia (15 of 18, 83.3%) with highly variable peak viral loads at T2 (11 of 18, 61.1%) or less frequent at T3 (four of 18, 22.2%) (the highest peak level was at 2.6x10^6^ HCMV DNA copies/mL) (Figure 1). Only one mother (one of 18, 5.6%) did not reactivate at any of the four time ranges, two mothers (two of 18, 11.1%) showed only a slight increase of viral load at T4.

In contrast, corresponding plasma samples of all mothers at all time ranges (72 of 72, 100%) were negative for HCMV DNA using nested PCR. 

### 3.2. HCMV Specific-IgG-Antibodies in Plasma and Whey Samples

HCMV-specific-IgG and -IgM in plasma from a cohort of 18 seropositive mothers of the BlooMil study were evaluated with electrochemiluminescence immunoassay (ECLIA). All mothers were latently infected with high IgG levels and high avidity and in all but one instance, without IgM detection. This one mother showed low persistent IgM reactivity.

In total, the plasma ECLIA IgG readouts of the mothers ranged from 40 to 4532 U/mL (Figure 2A). An increase of 227 U/mL of the mean level from T1 (10–15 days pp, 604 U/mL) to T4 (55–60 days pp, 831 U/mL) was observed. A Friedman test showed significant differences between plasma IgG levels (*p* < 0.001) over all four time ranges. The post hoc test showed significant differences between T1 and T3 (*p* = 0.003), as well as between T1 and T4 (*p* < 0.001). 

In milk whey, HCMV-specific-IgG levels (ECLIA) were detectable only in 13 of 18 (72.2%, including borderline reactivity three of 18) mothers (Figure 2B). In five of 18 (27.8%) mothers, ECLIA assay results were under the detection limit. The ECLIA values in whey ranged from cut-off level 1 to 42 U/mL. An increase of HCMV-specific-IgG in whey was also observed during viral reactivation in most of the samples. Only slight or no increases were detected, especially in HCMV-IgG borderline samples. Friedman test results showed no significant differences over all four time ranges; however, comparing single time ranges with Wilcoxon matched-pairs signed rank test without corrections for multiple testing resulted in a significant increase from T2 (mean level of 2.9 U/mL) to T4 (mean level of 7.8 U/mL) (*p* = 0.04, Figure 2B). The mean HCMV-specific-IgG levels in whey increased, especially after peak viral load (Figure 1), resulting in a 2.25-fold increase from T2 to T3. 

The plasma IgG mean levels were 275-fold higher than milk whey IgG mean levels. At T1 during early viral reactivation, a correlation of plasma to whey IgG’s was detected (Spearman’s rho 0.75, *p* = 0.004), while the correlation was lower at T4 (Spearman’s rho 0.57, *p* = 0.045) (Figure 2C,D). 

Commercially available recombinant antigen IgG immunoblots of plasma and whey samples from seropositive mothers were performed to detect HCMV antigen-specific antibodies during viral reactivation. The available antigens are of relevance for primary- and re-infection with HCMV and were used to gain information about specific antibody-patterns during reactivation. In plasma, antibodies against IE1 and CM2 showed fluctuating reactivities with a slight increase to the later time range using a 100-fold dilution as recommended by the manufacturer (Figure 3). Against both of these proteins, an IgG-seroconversion was detectable in three mothers from T1 to T3/T4 (Figure 3A–D). 

In contrast, only antibodies against the phosphoprotein p150 in plasma were continuously expressed from all mothers (18 of 18, 100%) with a very high reactivity (Figure 4A,B). Anti-p65, like anti-IE1 and anti-CM2-antibodies, were fluctuating in their reactivities amongst the individual mothers (Figure 4C,D). 

Antibodies against gB region 1 and 2 were also variably expressed (Figure 5). However, anti-gB1 antibodies were detectable in all plasma samples (18 of 18, 100%). Anti-gB2-IgG-reactivity in plasma did not change during viral reactivation and was reactive in 15 of 18 (83.3%) women (Figure 5C,D).

Based on prior experiments, milk whey samples were only diluted two-fold to be able to detect HCMV-specific antibodies in recomLine blots. A slight increase from T1 to T3/T4 in detectable antibody reactivities against all of the six recombinant antigens was observed (Figure 3, Figure 4 and Figure 5). Five mothers developed antibody-reactivity in mature milk against IE1 based on reduction of non-reactive cases (Figure 3A,B). Only anti-p150-IgG was continuously expressed in all whey samples (Figure 4A,B), but with distinctly lower reactivities than in plasma. Anti-gB1-IgG, which, like anti-p150-IgG, was also continuously expressed in plasma (18 of 18, 100%), was only reactive in five of 18 (27.8%) mothers’ whey at T1. Three additional mothers developed reactivity in T3/T4 (Figure 5A,B). Anti-gB2 reactivity was also only detectable in five of 18 (27.8%) mothers.

### 3.3. Neutralization Assays

Plasma and whey pools of T1 and T4 from all the HCMV-seropositive and -seronegative BlooMil study mothers were generated. All pools were used for detection of neutralizing antibodies. For calibration of seropositive samples, HIG-stock solution was diluted to the HCMV-IgG level of the seropositive cohort. The seronegative T1 and T4 plasma pools showed high numbers of HCMV IEA-plaques (Figure 6A) reproducible in all dilutions. The seropositive plasma pools showed a decrease of plaque counts and therefore, a higher NT-capacity (Figure 6C), from T1 with calculated NT-50 value of 1:3000 to T4 with calculated NT-50 value of 1:4000. A NT-capacity of 100% in plasma was observed using dilutions up to 800-fold. As shown in Figure 6C, the increase of the NT-capacity between T1 and T4 can be detected at dilutions of 1:1600 and 1:3200. The calibration of the HIG preparation using mean HCMV-IgG ECLIA levels of all seropositive plasma samples appeared to be very suitable, since the IEA plaque counts ranged exactly between those of the T1 and T4 pools.

When the HIG normalization to mean ECLIA values of milk whey was used, intrinsic HIG neutralization was not detectable in the dilutions at or above 1:256 (Figure 6B). Based on the low HCMV-specific-IgG-antibody concentration in whey, the impact of HIG predilution on neutralizing activity was only seen in lower dilution steps (1:32, 1:128). In contrast, even milk whey from HCMV-seronegative mothers was capable of neutralizing HCMV, which indicates the presence of competing unspecific neutralizing components. Therefore, the seronegative pools for each dilution were set as 100% reference for the calculation of the NT-capacity of each of the HCMV-seropositive pools (Figure 6D). 

The calculated NT-50 values in whey supported the results of the increased ECLIA values with a slightly higher NT-capacity from T1 with 1:86 to T4 with 1:100.

## 4. Discussion

The frequency of viral reactivation in breast milk of HCMV-IgG-positive mothers is nearly equal to their serostatus [30]. The onset of viral reactivation occurs early, mostly already during the first two weeks after birth. The mechanism of the viral reactivation during lactation is still unknown. Viral shedding into breast milk is self-limited and usually ends two to three months after birth. Two immune control pathways may be responsible for this phenomenon: the humoral immunity with virus-specific antibodies and the cellular immunity with HCMV-specific T cells, NK cells and other immune cell subpopulations.

In this study, we investigated HCMV-specific antibody kinetics during viral reactivation in breast milk. Almost all the mothers in our study cohort showed unimodal viral load courses in whey with very high variations of peak levels, which confirms earlier reports [2,31]. Only one mother participating in the BlooMil study did not reactivate the virus (one of 18, 5.6%) at the given time ranges, which is consistent with previously found reactivation rates [30]. Overall, the onset of HCMV-reactivation and the discontinuation of viral shedding in breast milk were clearly observable during the chosen time ranges until two months after birth.

During pregnancy, IgG-antibodies are transferred via the neonatal Fc receptor to the fetal blood stream [32,33]. In total, 75% of the HCMV-specific-IgG concentrations measured in umbilical cord blood were even higher than those in maternal sera [34]. This might be attributable to the decrease of total IgG concentrations found in maternal serum, especially during the second and third trimester [35,36,37], when the most IgGs are transferred to the fetus. Moreover, the plasma volume of pregnant women increases by up to one liter [38], which could also contribute to lowering total IgG levels during pregnancy. Therefore, the increase of plasma HCMV-specific-IgG ECLIA-levels after birth in the seropositive cohort in this study with a mean increase of 227 U/mL from T1 to T4 might be the result of IgG levels returning to pre-pregnancy levels. On the other hand, the HCMV-specific-IgG increase in plasma could be generally indicative of a systemic virus reactivation. However, in our study, all individual plasma samples of breastfeeding mothers during all four time ranges were negative for HCMV DNA and therefore, a systemic infection could be excluded. It has already been shown that HCMV-reactivation is a local process in the breast [31,39,40]. Consistently with this local reactivation, all mothers showed a complete lack of a disseminated HCMV-infection in the absence of viral DNAemia, while they shedded virus into breast milk. Furthermore, these mothers showed serological characteristics of a latent HCMV infection with high HCMV-IgG levels and high IgG-avidity and lack of IgM detection. Azenkot et al. [41] found positive samples in a minority of vaginal secretions (three of nine mothers) and saliva (two of nine) but without viral DNAemia detection. 

Interestingly, in our study, whey HCMV-specific-IgGs showed an increase after peak viral load. The correlation between plasma and milk whey IgGs decreased from T1 to T4, which underlined a relative unproportional IgG-elevation in whey.

Although 100% of mothers continuously expressed strong anti-p150 and variably anti-gB1 IgG-antibodies in plasma, the antibody profile varied from mother to mother. Anti-gB2 reactivity was at 83.3% of all seropositive mothers in our study. This confirms previous data which demonstrated that approximately 82% of seropositive individuals express anti-gB2 antibodies [22,25]. The NT-capacity of plasma increased from T1 to T4 (NT-50-values 1:3000 and 1:4000 respectively), which might reflect an increase of total IgG’s in plasma. 

In milk whey, only 72.2% of the seropositive mothers of our study showed HCMV-specific antibodies. In contrast, Kassim et al. [42] found 20% of HCMV-specific antibody positive breast milk samples. When recomLine blots with a two-fold dilution for milk whey were used, all samples were positive at least for anti-p150-IgG-antibodies. This discrepancy might be due to the sensitivities of test systems and recombinant antigens used. 

The IgG-seronegative whey pool showed a high unspecific neutralizing capacity. This might be due to milk oligosaccharides, which were shown to be highly abundant in milk and inhibit different intestinal and respiratory viruses [43] and might, therefore, also be able to interfere with the neutralization test system. Alternatively, sIgA might also influence the neutralization test system, but detailed data on the role of HCMV-specific-sIgA in epithelial based NT-assays are not available according to our knowledge. Moreover, lactoferrin [44], which can prevent cell entry of HCMV [45] and other proteins like lysozyme and lactoperoxidase [46], might play an important role of unspecifically inhibiting substances in whey. 

The assay calibration with prediluted HIG preparation to whey HCMV-IgG concentrations [29] showed NT-capacities up to 256-fold dilutions. Still, in whey, an increase of the plaque counts was observable over all dilution steps, which, again, underlined the role of unspecifically neutralizing milk whey components.

Anti-gB IgG is thought to be important for the protection against HCMV infections in infants and has a high neutralizing capacity [25,47]. In our study, the anti-gB1 and 2 IgG’s found in milk whey were only weakly expressed and only slight increases were detectable over time. Still, our results showed increased NT-capacity in T4 compared to T1 (over twice). This effect, and also the higher plaque counts in T4, showed that the unspecific neutralizing substances in T4 are less dominant than in T1, which is still temporally close to colostrum which is known to have high concentrations of immunologically active ingredients [48].

Like Ehlinger et al. [49], we could not find a correlation between viral load and IgG titers. However, the whey viral load still seemed to play an important role in our study since IgG-antibodies increased after peak viral load.

Unexpectedly, neutralization was very low in breast milk compared to plasma, when the seronegative pool with its unspecific neutralizing capacity was subtracted (NT-50 values of 1:100 to 1:4000 at T4). Ehlinger et al. [49] reported similar results. One reason for the low ECLIA-IgG values and NT-capacity in breast milk (mean plasma/whey quotient of 275-fold) could be that endogenous HCMV is present in milk [50] (with up to 2.6x10^6^ copies/mL in our study) and some antibodies might already have bound to the virus and can therefore not be measured or neutralize additional exogenous virus. As published earlier, the low HCMV-specific IgG in whey also reflects an overall lower IgG concentration in whey compared to plasma [49]. HCMV-specific IgG and IgA seem to be slightly higher in breast milk supernatant compared to plasma, but only when normalized to total IgG or IgA concentrations [49]. 

A limitation of our study was the overall low number of mothers of preterm infants, their low milk volumes, and their critical attitude against additional blood donations after birth. This resulted in a low number of study participants. Another limitation was that we were not able to document the mother–infant transmission due to exclusive feeding of short-term pasteurized breast milk [28]. Samples from saliva, urine or vaginal swabs were not collected, since the main goal was to describe humoral immune response during viral reactivation during lactation. This study mainly focusses on IgG; therefore, another limiting factor is the unknown influence of highly abundant sIgA in whey.

To our knowledge, this study showed the first HCMV-specific-IgG kinetics in breast milk. The increase from T1 to T4 could have an effect on the HCMV load decrease in whey, but the primary cause might be found elsewhere. Therefore, our analysis of T cell subpopulations and their potential role in the decrease of viral load is continuing. 

## Figures and Tables

**Figure 1 nutrients-12-00338-f001:**
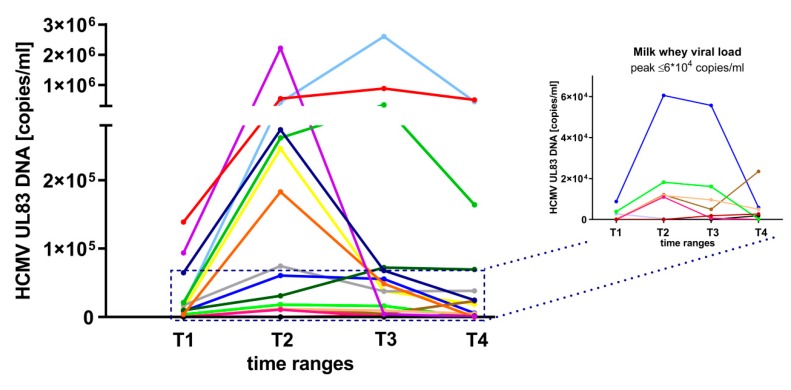
Milk whey viral loads (target region UL83) of 17 of 18 HCMV-seropositive BlooMil study mothers at T1 (10 to 15 days pp), T2 (25 to 30 days pp), T3 (40 to 45 days pp) and T4 (55 to 60 days pp). The insert shows low viral loads (peak levels ≤ 6x10^4^ copies/mL). One mother did not reactivate during observation. Individual color identification.

**Figure 2 nutrients-12-00338-f002:**
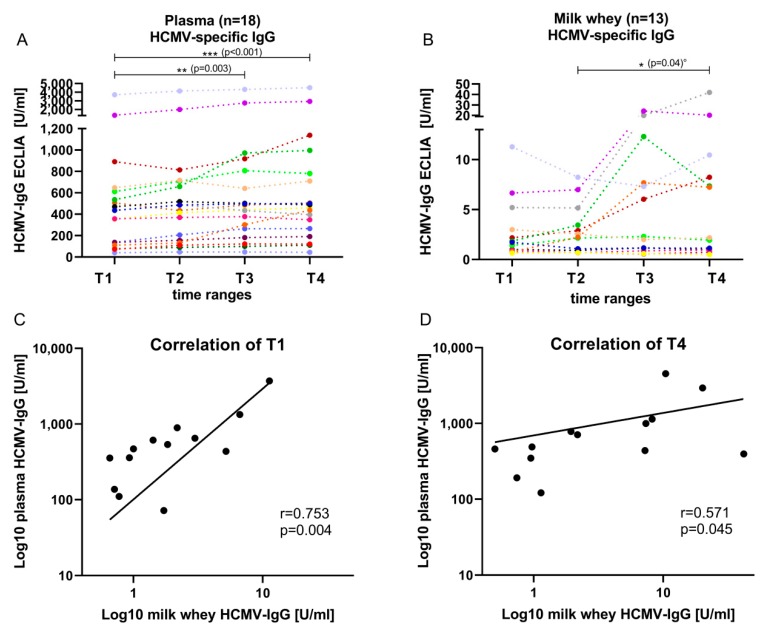
HCMV-specific-IgG ECLIA values (mean + 95%CI) of 18 seropositive mothers of the BlooMil study in (**A**) plasma and (**B**) milk whey from T1 (10 to 15 days pp), T2 (25 to 30 days pp), T3 (40 to 45 days pp) and T4 (55 to 60 days pp). The Friedman test was used for the kinetics and showed a significant change in plasma from T1 to T4 *p* < 0.001 and T1 to T3 *p* = 0.003, but in whey, no significant change was detected. °The Wilcoxon matched-pairs signed rank test was used in whey to compare single time ranges; no correction was used. Individual color identification as in Figure 1. Correlation of plasma and milk whey HCMV-specific-IgG were determined by Spearman’s rank correlation and non-linear regression at (**C**) T1 with r = 0.75, *p* = 0.004 and (**D**) T4 with r = 0.57, *p* = 0.045.

**Figure 3 nutrients-12-00338-f003:**
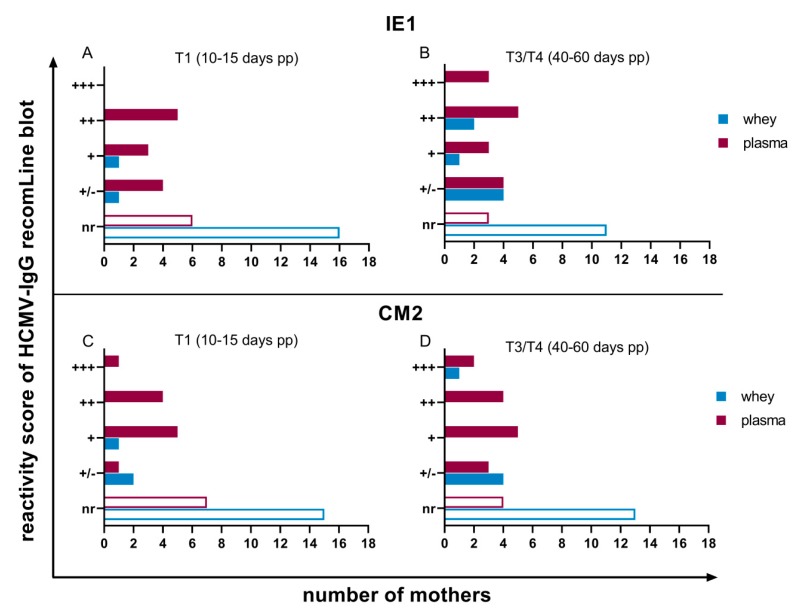
Reactivity scores of IgG recomLine blots for anti-recIE1-IgG at (**A**) T1 (10 to 15 days pp) and (**B**) T3/T4 (40 to 60 days pp) and anti-recCM2-IgG at (**C**) T1 (10 to 15 days pp) and (**D**) T3/T4 (40 to 60 days pp). The X-axis shows the number of study participants. Nr: non-reactive, +/-: lower as cut-off, but detectable, +: one-fold, ++: two-fold, +++: three-fold stronger than cut-off reactivity. Breast milk was diluted two-fold, plasma 100-fold.

**Figure 4 nutrients-12-00338-f004:**
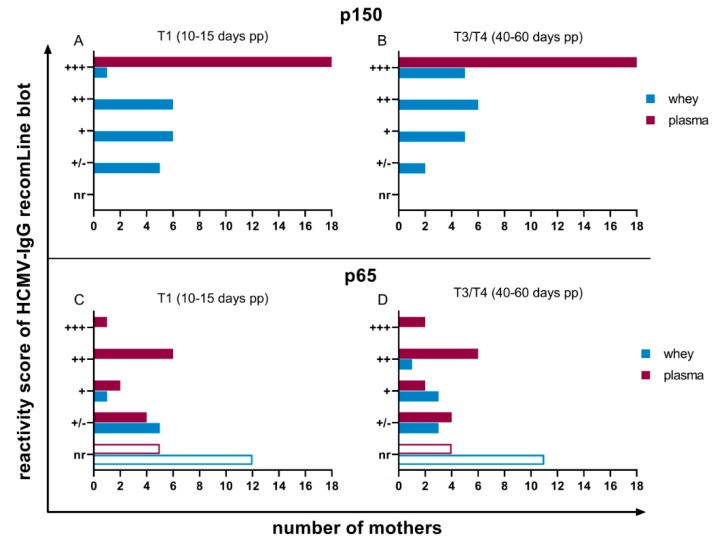
Reactivity scores of IgG recomLine blots for anti-rec150-IgG at (**A**) T1 (10 to 15 days pp) and (**B**) T3/T4 (40 to 60 days pp) and anti-recp65-IgG at (**C**) T1 (10 to 15 days pp) and (**D**) T3/T4 (40 to 60 days pp). Scoring of antibody reactivity as in Figure 3.

**Figure 5 nutrients-12-00338-f005:**
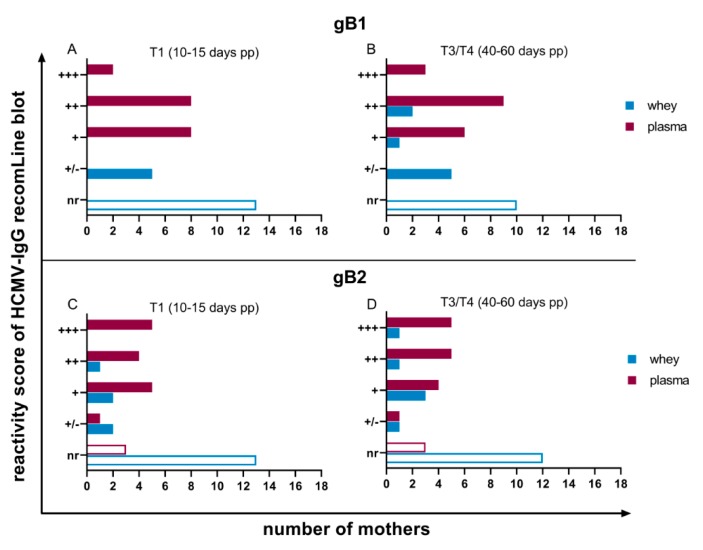
Reactivity scores of IgG recomLine blots for anti-recgB1-IgG at (**A**) T1 (10 to 15 days pp) and (**B**) T3/T4 (40 to 60 days pp) and anti-recgB2 at (**C**) T1 (10 to 15 days pp) and (**D**) T3/T4 (40 to 60 days pp). Scoring of antibody reactivity as in Figure 3.

**Figure 6 nutrients-12-00338-f006:**
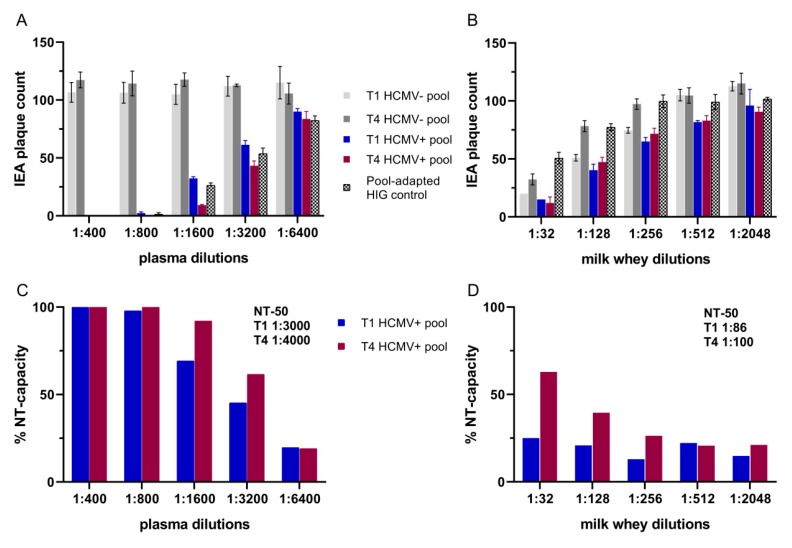
Neutralization assays of HCMV-seropositive and -negative pools using ARPE-19 target cells. (**A**) Plasma and (**B**) whey immediate early antigen (IEA) plaque counts in a two-fold serial dilution (mean+SD). HCMV-IgG calibration using hyperimmunoglobulin (HIG) preparation (prediluted 1:700 to mean ECLIA values of whey and 1:6.5 for plasma values). Neutralization (NT)-capacity in (**C**) plasma and in (**D**) whey were calculated by using the HCMV-seronegative pools as 100% reference for every dilution step. NT-50 values were calculated with a Probit analysis of the seropositive pool with mean values of the seronegative pools as reference.

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
