# Peer review of "Human Cytomegalovirus Reactivation During Lactation: Impact of Antibody Kinetics and Neutralization in Blood and Breast Milk"

_nutrients, 2020, doi:10.3390/nu12020338_

Round 1

Reviewer 1 Report

In this study, Lazar and colleagues investigated the impact of neutralization antibody kinetics and neutralization in blood and breast in the context of HCMV reactivation during lactation. They described a nice study. The manuscript is generally well-written, although some editing would be recommended. At this stage of development, several aspects of the manuscript need to be improved.

1. Resolution of the figures should be improved.

2. The English throughout needs work.

Author Response

All figures are exported with 600 dpi and the problem reviewer 1 described might be based on the resolution of the pdf file.

All figures have high resolution quality.

The manuscript was spell-and grammar checked by a native speaker, which now appears also in the Acknowledgements (Norman Davis).

Reviewer 2 Report

This is an interesting study, that uses multiple techniques to look at anti-HCMV IgG levels, specificity, and neutralising activity, in breast milk. Surprisingly, it shows that such antibodies are very low level, and have very little effect on HCMV in this body compartment. This is a novel and interesting finding, however I do have a concern that this may simply reflect the fact that they have measured IgG, and not IgA, which is the predominant Ig in breast milk. I have a few comments that would improve the manuscript:

Major points:

Discussion – the authors need to mention that a major limitation is the fact that they only measured IgG, and not IgA. Their data also needs interpreting in light of this fact. E.g. what is the relative concentration of IgG in plasma compared to breast milk? Do they think that there is a specific loss of HCMV-specific IgG in breast milk, or are the low levels simply reflective of low overall IgG levels compared to plasma? What is known about HCMV IgA responses? Given the low neutralising capacity of the whey, what is known about the ability of HCMV to induce neutralising IgA?

Line 92 – all measurements were performed on whey, is there any evidence of antibodies being retained in the fat layer?

Line 108 – are ECLIA & recomLine validated for antibodies in whey? I.e. if Cytotect is spiked into whey, and these assays are performed, is it detected at the appropriate level?

Minor points:

It’s not clear how were mothers chosen – were these simply seropositive/negative mothers, or was there any other selection criteria?

Line 131 – I assume from the data that Cytotect was spiked into the seronegative plasma? If so, it needs to be stated here.

Line 202 – the wording here is hard to follow, where it says ‘reduction of initial non-reactive samples’. I would favour simply stating that in three mothers, HCMV-specific IgG was not detected at T1, but became detectable at T3/T4.

Line 257 – the phrase ‘dilution steps showed the full dynamic range of neutralizing activity’ is hard to understand. I think what the authors mean is that ‘even milk from HCMV seronegative mothers was capable of neutralising HCMV, indicating the presence of competing unspecific neutralizing components’.

Line 257 – how sure are the authors of their interpretations, given that HIG didn’t appear to work at all in this assay? What is their interpretation of this point? Similarly, is the increase in neutralising activity at T4 a genuine increase, or simply a reduction in non-specific neutralisation?

Figure 6 – the IgG concentrations by ECLIA were quite variable between patients, yet neutralisation by whey antibodies appears fairly uniform across patients? How do the authors interpret this? Is it simply that the non-specific effects of milk mask the specific neutralisation by antibody? Or is it that neutralisation is being mediated by IgA, which the ECLIA didn’t measure?

Line 296 states that the mothers shed virions in their breast milk, yet I don’t see the data. Is that data published? If not, how many mothers shed virions?

Line 318 appears to suggest that sIgA could be non-specifically inhibiting infection. Is there existing evidence that this can be the case? How would this occur?

Line 325 – as above, ‘the whole dynamic range was visible’ is a hard phrase to understand.

Line 340 – the authors comment that HCMV was present at 2.6x106 copies/ml – I assume this is genome? If so, genomes may not represent virions. Conversely, if they are measuring virions, then they are clearly not being neutralised.

Author Response

Review Report Form 2

Open Review

English language and style

( ) Extensive editing of English language and style required
(x) Moderate English changes required
( ) English language and style are fine/minor spell check required
( ) I don't feel qualified to judge about the English language and style

Yes

Can be improved

Must be improved

Not applicable

Does the introduction provide sufficient background and include all relevant references?

(x)

( )

( )

( )

Is the research design appropriate?

(x)

( )

( )

( )

Are the methods adequately described?

(x)

( )

( )

( )

Are the results clearly presented?

(x)

( )

( )

( )

Are the conclusions supported by the results?

( )

(x)

( )

( )

Comments and Suggestions for Authors

This is an interesting study, that uses multiple techniques to look at anti-HCMV IgG levels, specificity, and neutralising activity, in breast milk. Surprisingly, it shows that such antibodies are very low level, and have very little effect on HCMV in this body compartment. This is a novel and interesting finding, however I do have a concern that this may simply reflect the fact that they have measured IgG, and not IgA, which is the predominant Ig in breast milk. I have a few comments that would improve the manuscript:

Major points:

Discussion – the authors need to mention that a major limitation is the fact that they only measured IgG, and not IgA. Their data also needs interpreting in light of this fact. E.g. what is the relative concentration of IgG in plasma compared to breast milk? Do they think that there is a specific loss of HCMV-specific IgG in breast milk, or are the low levels simply reflective of low overall IgG levels compared to plasma? What is known about HCMV IgA responses? Given the low neutralising capacity of the whey, what is known about the ability of HCMV to induce neutralising IgA?

(Reviewers questions in italic)

Answer

Our focus was the study of HCMV-specific IgGs in breast milk versus plasma. We added the following text changes to the discussion (in yellow) :

Page11, line 366
As published earlier, the low HCMV-specific IgG levels in whey reflect an overall lower IgG concentration in whey compared to plasma [49]. HCMV-specific IgG and IgA seem to be slightly higher in breast milk supernatant compared to plasma, but only when normalized to total IgG or IgA concentrations [49].

Discussion limitation paragraph page 11, line 376:
This study mainly focusses on IgG, therefore, another limiting factor is the unknown influence of highly abundant sIgA in whey.

Line 92 – all measurements were performed on whey, is there any evidence of antibodies being retained in the fat layer?

Following two sequential (500g and 1780g) centrifugation steps, the creamy milk top layer contains strongly hydrophobic long fatty acids, triglycerides, sphingolipids, glycolipids and long as well as short chain fatty acids – either saturated or unsaturated. This layer does not contain greater amounts of hydrophilic antibodies (see milk separation protocol, Hamprecht, Witzel, 2003 [19]).

Line 108 – are ECLIA & recomLine validated for antibodies in whey? I.e. if Cytotect is spiked into whey, and these assays are performed, is it detected at the appropriate level?

We thank the reviewer for this important question:
We performed validation experiments and spiked whey with Cytotect (Cytotect’s end concentration at 1:100). We got a recovery rate of 88.3%. The same was done for recomLine blots and antibodies against all antigens were detectable. We added the following to the methods section:

ECLIA, page 3, line 110:
For validation of whey, a spike and recovery experiment for the HMCV-IgG ECLIA test system was performed. Therefore, whey and the PBS controls were spiked with a 100-fold diluted Cytotect® stock solution. The recovery rate in the spiked whey sample was at 88.3%.

RecomLine blot, page 3, line 132:
A validation experiment was performed, using hyperimmunoglobulin (HIG: Cytotect®, Biotest® Pharma; charge B797033; 50mg/ml)-spiked whey. Antibodies against all antigens were detectable without any influence on reactivity against the phosphoproteins p65 and p150. Only a very slight decrease in reactivity was observed against the recombinant antigens gB1 and gB2, IE1 and CM2. This did not interfere with our antibody analysis in milk whey.

Minor points:

It’s not clear how were mothers chosen – were these simply seropositive/negative mothers, or was there any other selection criteria?

Healthy, breastfeeding mothers of mostly preterm infants were included in our study. HCMV reactivations and antibody patterns of seropositive mothers of preterm and term infants showed no differences in previous studies (Hamprecht, Goelz 2017, Ehlinger 2011).

We added to the text the following: page 2, line 78:

36 healthy and immunocompetent mothers with known HCMV-serostatus participated, 8 had to be excluded from the analysis due to lack of breast milk or transfer to a peripheral hospital.

Line 131 – I assume from the data that Cytotect was spiked into the seronegative plasma? If so, it needs to be stated here.

Cytotect was spiked into medium. We added it to the methods section page 4, line 146:

The mean ECLIA values of the seropositive plasma (720 U/ml) and whey samples (6.4 U/ml) were calibrated by predilution of Cytotect® (4725 U/ml) [29] in medium to both corresponding concentrations (1:6.5 and 1:700 respectively)

Line 202 – the wording here is hard to follow, where it says ‘reduction of initial non-reactive samples’. I would favour simply stating that in three mothers, HCMV-specific IgG was not detected at T1, but became detectable at T3/T4.

We agree with the reviewers proposal and changed the wording, page 6, line 220:
Against both of these proteins an IgG-seroconversion was detectable in three mothers from T1 to T3/T4 (Figure 3A,B and C,D).

Line 257 – the phrase ‘dilution steps showed the full dynamic range of neutralizing activity’ is hard to understand. I think what the authors mean is that ‘even milk from HCMV seronegative mothers was capable of neutralising HCMV, indicating the presence of competing unspecific neutralizing components’.

Yes, we agree and changed the text into the wording of the reviewer. Page 9, line 279

Line 257 – how sure are the authors of their interpretations, given that HIG didn’t appear to work at all in this assay? What is their interpretation of this point? Similarly, is the increase in neutralising activity at T4 a genuine increase, or simply a reduction in non-specific neutralisation?

We think there might be a misunderstanding:
We think HIG worked nicely as a positive control for HCMV-specific neutralization. It was prediluted to the same amount of antibodies in breast milk measured with ECLIA and since their levels are low, it was expected that the dilution factors are too high to actually observe still neutralizing capacities. The interesting part is that in breast milk, we still see neutralizing activity in every dilution step, what underlines the unspecific background neutralization of whey. 
Yes, we assume the increase in neutralising activity at T4 is a genuine increase, since it was normalized to the corresponding unspecific neutralization effects of the respective T1 (early, close to colostrum) and T4 (mature milk) time ranges of the seronegative cohorts.

Figure 6 – the IgG concentrations by ECLIA were quite variable between patients, yet neutralisation by whey antibodies appears fairly uniform across patients? How do the authors interpret this? Is it simply that the non-specific effects of milk mask the specific neutralisation by antibody? Or is it that neutralisation is being mediated by IgA, which the ECLIA didn’t measure?

We think there was a misunderstanding:
We generated 8 pools of plasma and milk whey as shown in Figure 6. Samples were pooled to get either whey or plasma pools of all seropositive or seronegative mothers at T1 and T4. The error bars indicate results from triplicates and do not reflect interindividual variations.

Line 296 states that the mothers shed virions in their breast milk, yet I don’t see the data. Is that data published? If not, how many mothers shed virions?

We also have microculture results of whey, but the data are not shown here in this manuscript. We changed the wording ‘virion’ into virus. Page 10, line 317

Line 318 appears to suggest that sIgA could be non-specifically inhibiting infection. Is there existing evidence that this can be the case? How would this occur?

We did not find any evidence that sIgA inhibited the in vitro CMV-infection of epithelial target cells in our NT-assay. Nevertheless, we did not determine sIgA levels in milk. In general, highly abundant total sIgA levels might be able to interfere unspecifically with our NT assay results.

We changed the text to the following  page 10, line 340:
Alternatively, sIgA might also influence the neutralization test system but detailed data on the role of HCMV-specific-sIgA in epithelial based NT-assays are not available according to our knowledge.

Line 325 – as above, ‘the whole dynamic range was visible’ is a hard phrase to understand.

We changed the wording into: (page 10, line 347)
Still, in whey an increase of the plaque counts was observable over all dilution steps.

Line 340 – the authors comment that HCMV was present at 2.6x106 copies/ml – I assume this is genome? If so, genomes may not represent virions. Conversely, if they are measuring virions, then they are clearly not being neutralised.

The semi quantitative microculture assay (number of CMV IE pos fibroblast nuclei/20 000 target cells per well) and PCR results (in genome equivalents/ml of milk whey) correlate quite well [2], but since we did not show these data we made the following changes in the text page 11, line 362:

One reason for the low ECLIA-IgG values and NT-capacity in breast milk (mean plasma/whey quotient of 275-fold) could be that endogenous HCMV is present in milk [50] (with up to 2.6x106 copies/ml in our study) and some antibodies might already have bound to the virus and can therefore not be measured or neutralize additional exogenous virus.

Submission Date

12 December 2019